# Holy Basil (*Ocimum sanctum* L.) Flower and Fenofibrate Improve Lipid Profiles in Rats with Metabolic Dysfunction Associated Steatotic Liver Disease (MASLD): The Role of Choline Metabolism

**DOI:** 10.3390/plants14010013

**Published:** 2024-12-24

**Authors:** Siraphat Taesuwan, Jakkapong Inchai, Konpong Boonyingsathit, Chanika Chimkerd, Kunchit Judprasong, Pornchai Rachtanapun, Chatchai Muanprasat, Chutima S. Vaddhanaphuti

**Affiliations:** 1Center of Excellence in Agro Bio-Circular-Green Industry (Agro BCG), Faculty of Agro-Industry, Chiang Mai University, Chiang Mai 50100, Thailand; siraphat.t@cmu.ac.th (S.T.); pornchai.r@cmu.ac.th (P.R.); 2Functional Foods and Nutrition Research (FFNR) Laboratory, University of Canberra, Bruce, Canberra, ACT 2617, Australia; 3Innovative Research Unit of Epithelial Transport and Regulation (iETR), Department of Physiology, Faculty of Medicine, Chiang Mai University, Chiang Mai 50200, Thailand; jakkapong.inc@gmail.com; 4Institute of Nutrition, Mahidol University, Nakhon Pathom 73170, Thailand; konpong.b@gmail.com (K.B.); kunchit.jud@mahidol.ac.th (K.J.); 5Faculty of Pharmacy, Mahidol University, Bangkok 10400, Thailand; chanika.chimkerd@gmail.com; 6Chakri Naruebodindra Medical Institute, Faculty of Medicine Ramathibodi Hospital, Mahidol University, Samut Prakan 10540, Thailand

**Keywords:** choline, holy basil, metabolic syndrome, non-alcoholic fatty liver disease, obesity, one-carbon metabolism, polyphenols

## Abstract

Metabolic dysfunction-associated steatotic liver disease (MASLD) is linked to choline metabolism. The present study investigated the effect of holy basil (*Ocimum sanctum* L.) flower water extract (OSLY) on MASLD with choline metabolism as an underlying mechanism. Rats with high-fat diet (HFD)-induced MASLD received 250–1000 mg/kg bw of OSLY, fenofibrate, or fenofibrate + 1000 mg/kg OSLY combination. Biochemical parameters, choline metabolites, and one-carbon gene transcription were analyzed. OSLY and fenofibrate independently reduced serum LDL cholesterol (*p* < 0.02), liver cholesterol (*p* < 0.001), and liver triglyceride levels (*p* < 0.001) in HFD-fed rats. Only OSLY reduced signs of liver injury and increased serum HDL. Fenofibrate influenced choline metabolism by decreasing liver glycerophosphocholine (GPC; *p* = 0.04), as well as increasing betaine (*p* < 0.001) and the betaine:choline ratio (*p* = 0.02) in HFD-fed rats. Fenofibrate (vs. HFD) increased the expression of one-carbon metabolism genes *Mthfd1l*, *Pemt*, *Smpd3*, and *Chka* (*p* < 0.04). The OSLY treatment decreased liver GPC (500 mg dose; *p* = 0.03) and increased *Smpd3* expression (1000 mg dose; *p* = 0.04). OSLY and fenofibrate showed weak synergistic effects on lipid and choline metabolism. Collectively, OSLY and fenofibrate independently improve lipid profiles in MASLD rats. The benefits of fenofibrate are partially mediated by choline/one-carbon metabolism, while those of OSLY are not mediated by this pathway. Holy basil flower extract merits further development as an alternative medicine for MASLD.

## 1. Introduction

Metabolic dysfunction-associated steatotic liver disease (MASLD), formerly known as non-alcoholic fatty liver disease (NAFLD), is a rapidly increasing pandemic with a current estimated global prevalence of 38% [1]. Insulin resistance, a key pathology of MASLD, disrupts hepatic glucose regulation, promotes gluconeogenesis and de novo lipogenesis, and subsequently leads to extrahepatic hyperglycemia and hypertriglyceridemia [2]. Hepatic lipogenesis is mediated by the phosphoinositide-3-phosphate kinase/Akt signaling pathway. Upon insulin stimulation, Akt activates the mechanistic target of rapamycin complex 1 (mTORC1), which in turn activates sterol regulatory element-binding protein 1c (SREBP1c), a key transcription factor in lipogenesis, causing hepatic steatosis [2,3,4].

Choline is a vitamin B-like nutrient that is essential in hepatic lipid metabolism. Choline is a precursor of phosphatidylcholine (PC)—a key phospholipid required for the biosynthesis and export of VLDL from the liver [5]. Long-term inadequate choline intake was shown to cause NAFLD, as well as liver and muscle damage [5]. Choline also acts as a methyl donor in one-carbon metabolism. Its derivative, betaine, provides methyl groups for the production of methionine and subsequently *S*-adenosylmethionine, a universal methyl donor for cellular methylation reactions [6]. A feedback mechanism exists between choline metabolism and SREBP1, whereby SREBP1 controls the production of *S*-adenosylmethionine and PC [7]. The blocking of PC production causes elevated SREBP1 and lipid accumulation in mouse liver [7]. A PC species has also been identified as an endogenous agonist of the peroxisome proliferator-activated receptor alpha (PPARα) [8], a nuclear receptor that can induce the transcription of genes involved in fatty acid oxidation, ketogenesis, lipid transport, and gluconeogenesis [9,10]. Portal vein infusion of this PC resulted in decreased hepatic steatosis in mice [8]. Since PC acts as a PPARα activator, it may enhance the effects of fenofibrate, a known PPARα activator and an FDA-approved drug [11].

Holy basil (*Ocimum sanctum* L.) is an herb in the family Lamiaceae found ubiquitously in Southeast Asia and worldwide. Basil contains phenolic compounds such as chlorogenic acid, caffeic acid, and vanillic acid, as well as flavonoids such as luercetin, luteolin, rutin, apigenin, and kaempferol [12]. These compounds have been shown to increase PPARα (*Ppara*) gene expression and suppress *Srebp1c* in mouse liver. Furthermore, holy basil parts have shown anti-cancer, antioxidative, anti-inflammatory, and antidiabetic effects in vitro [12]. Holy basil has the potential to be developed into medications or supplements aimed at preventing metabolic diseases. However, previous studies primarily focused on the bioactivities of holy basil leaves, while their flower by-products may contain potent bioactivities that have not been explored.

Holy basil extracts have demonstrated the ability to reduce hepatic fat content and lower liver lipid peroxidation in a rodent model of hypercholesterolemia, possibly by acting through the SREBP1 pathway [13,14,15]. Due to the crosstalk between choline metabolism and the regulation of lipogenesis via mTOR/SREBP1, the choline pathway may mediate the effects of holy basil extracts on MASLD. This study aimed to investigate whether (1) *Ocimum sanctum* L. flower extracts (OSLY) altered choline metabolites and one-carbon metabolism gene expression in rats, (2) the changes in choline metabolism were correlated with changes in metabolic markers, and (3) choline metabolites mediated the effects of OSLY on metabolic markers. This study contributes to the field of plant science by characterizing the metabolites in the holy basil flower, a lesser-studied part of this common plant. Our research enhances the understanding of holy basil’s ecological contributions and highlights its potential as a source of active compounds for treating MASLD.

## 2. Results

### 2.1. Holy Basil Flower and Fenofibrate Reduced Hepatic Lipid Accumulation and Markers of Liver Injury and Oxidative Stress

Rats fed a high-fat diet (HFD) exhibited increased caloric intake, greater body weight gain, higher visceral fat levels, and larger liver weights compared to those fed a normal diet (ND) (Appendix A). Rats given an HFD and treated with fenofibrate gained less weight and had lower visceral fat than the HFD group, while liver weight and index remained unchanged. Fenofibrate increased kidney size and index, consistent with its reported side effects on renal functions. Administration of OSLY alone did not change caloric intake, body weight, or organ weights compared to the HFD group (*p* > 0.9).

HFD resulted in increased serum total and LDL cholesterol levels, as well as higher hepatic cholesterol and triglyceride content relative to ND, indicating hepatic steatosis (Appendix A). Administering OSLY or fenofibrate lowered serum total and LDL cholesterol levels, and hepatic cholesterol and triglyceride levels compared to the HFD group. However, no synergistic effects were observed between the two treatments. In addition, the 1000 mg/kg OSLY significantly elevated serum HDL cholesterol, reduced alanine transaminase levels, and lowered plasma malondialdehyde levels compared to the HFD group (Appendix A). Conversely, fenofibrate did not have any effect on these markers. The OSLY did not exhibit dose-dependent effects on any biochemical markers.

### 2.2. Holy Basil Flower and Fenofibrate Altered Hepatic Choline Metabolite and One-Carbon Metabolism Gene Expression Levels

#### 2.2.1. Hepatic Choline Metabolites

Hepatic choline metabolites responded to diet, OSLY, and fenofibrate treatments. In particular, HFD resulted in higher hepatic concentrations of choline, phosphocholine, and glycerophosphocholine (GPC), whereas the betaine:choline ratio was lower compared to the ND group (Figure 1). In HFD-fed rats, fenofibrate increased betaine concentrations and the betaine:choline ratio, while decreasing GPC concentrations. OSLY had minimal effects on choline metabolites, except for GPC. Neither OSLY nor fenofibrate prevented the rise in hepatic choline concentrations.

HFD did not significantly alter lipid-soluble hepatic choline metabolite concentrations compared to ND, although the PC concentration was visually decreased (Figure 2). Fenofibrate and OSLY did not exhibit a clear pattern that allowed for clear conclusions.

#### 2.2.2. One-Carbon Metabolism Gene Expression

One-carbon metabolism is a pathway involving enzymes that influence choline metabolites in the liver. This study measured the mRNA levels of key genes related to one-carbon metabolism, focusing on three pathways: the cytidine diphosphate (CDP)-choline pathway, the methionine cycle, and the folate-mediated pathways. The CDP-choline pathway converts dietary choline to PC and is responsible for 70% of total choline within the body. Within this pathway, choline kinase alpha (CHKA) and sphingomyelin phosphodiesterase 3 (SMPD3) convert free choline and sphingomyelin to phosphocholine. Next, phosphate cytidylyltransferase 1 choline α isoform (PCYT1A) converts phosphocholine to CDP-choline in the rate-limiting step of the CDP-choline pathway [16]. Finally, choline-ethanolamine phosphotransferase-1 (CEPT1) catalyzes the formation of PC.

HFD-fed rats had lower hepatic *Smpd3* mRNA levels than ND-fed rats (Figure 3). Administration of 1000 mg/kg of OSLY, fenofibrate, or their combination restored *Smpd3* expression, suggesting that the sphingomyelin→phosphocholine reaction was responsive to dietary manipulations. *Pcyt1a* expression did not differ among groups. *Cept1* expression increased only after administering the combination of fenofibrate and OSLY, suggesting a potential synergistic effect.

The methionine cycle generates methyl groups for cellular methylation. Choline, through its derivative betaine, donates a methyl group to form methionine via betaine-homocysteine methyltransferase (BHMT). Methionine is then converted to *S*-adenosylmethionine by methionine adenosyltransferase 1A (MAT1A), which is later demethylated by phosphatidylethanolamine *N*-methyltransferase (PEMT) to generate PC. The PEMT-derived PC is an endogenous choline product, accounting for the other 30% of choline in the body. In the current study, *Bhmt* mRNA levels did not differ across groups, while *Mat1a* mRNA levels were lower only in the HFD group compared to the ND group (Figure 4). The expression of *Pemt* was significantly increased by fenofibrate and its combination with OSLY, compared to the HFD control group. In contrast, OSLY alone did not produce the same effect. These findings suggest that fenofibrate, rather than OSLY, promotes the endogenous synthesis of PC in the liver.

In addition to choline, methyl groups are generated via folate-mediated one-carbon metabolism. The cytosolic form of methylenetetrahydrofolate dehydrogenase 1 (MTHFD1) and the MTHFD1−like mitochondrial form (MTHFD1L) are tri-functional enzymes responsible for generating folate that transports a methyl group (formate) to the methionine cycle. Figure 5 shows that *Mthfd1* and *Mthfd1l* were upregulated in the combination treatment compared to the HFD group. The elevated *Mthfd1* expression could result from the combined impact of fenofibrate and OSLY, while the heightened *Mthfd1l* expression was probably caused by fenofibrate alone. Notably, changes in one-carbon gene expression following OSLY treatments were limited to the 1000 mg dosage, and no dose-dependent relationships were observed.

### 2.3. Correlations Among Hepatic Choline Metabolites, One-Carbon Metabolism Gene Expression, and Metabolic Markers

Hepatic choline metabolites were mostly uncorrelated, except for aqueous metabolites, e.g., phosphocholine and GPC (Pearson’s *r* = 0.94, *p* < 0.0001; Figure 6). Moderate correlations were observed between *Pemt* and *Mthfd1l* (*r* = 0.61, *p* < 0.0001), *Pcyt1a* and *Cept1* (*r* = 0.58, *p* < 0.0001), and *Pcyt1a* and *Bhmt* (*r* = 0.48, *p* = 0.003).

Serum total and LDL cholesterol were positively correlated with serum alanine transaminase and aspartate transaminase. Hepatic triglycerides showed positive correlations with markers of lipid metabolism (hepatic and serum cholesterol), liver injury (serum alanine transaminase and aspartate transaminase), and oxidative stress (plasma and hepatic malondialdehyde). The liver index and serum alkaline phosphatase were positively correlated with *Pemt* and *Mthfd1l* expression (*r* = 0.44–0.74, *p* < 0.05), as well as hepatic choline and betaine concentrations (*r* = 0.58–0.81, *p* < 0.002). Choline and betaine were also positively correlated with *Pemt* (*r* = 0.51, *p* = 0.006 for choline and *r* = 0.67, *p* < 0.0001 for betaine) and *Mthfd1l* (*r* = 0.60, *p* = 0.007 for choline and *r* = 0.51, *p* = 0.006 for betaine). These correlation patterns suggest the co-upregulation of folate-mediated methyl group production and endogenous PC synthesis under metabolic stress.

### 2.4. Choline Metabolites Were Related to Metabolic Markers in a Treatment-Dependent and Independent Manner

To further investigate how choline metabolites mediated the treatment effects on metabolic markers, principal component analysis was used to consolidate the choline metabolites into three principal components that cumulatively explained 82% of the variance (factor loading is shown in Appendix A; the principal component plot is shown in Appendix A). Mediation analysis was used to assess the impact of these principal components on each metabolic marker in the presence or absence of treatment group effects. The analysis revealed that the influence of choline metabolites on the metabolic markers was mostly driven by the treatments. However, principal component 3 remained negatively associated (*p* = 0.04) with serum alkaline phosphatase after adjusting for the treatment effect, indicating that principal component 3 was independently related to serum alkaline phosphatase. Principal component 3 was also negatively and independently associated with serum creatinine (*p* = 0.02) and positively associated with estimated glomerular filtration rate (*p* = 0.01). The results indicate that a higher score for principal component 3—characterized by lower concentrations of hepatic betaine and choline, along with higher concentrations of sphingomyelin (see Appendix A)—may demonstrate better renal function. This improved renal function is evidenced by lower serum levels of alkaline phosphatase and creatinine, as well as an increased estimated glomerular filtration rate.

## 3. Discussion

### 3.1. High-Fat Diet Altered Hepatic Choline Metabolism in the Pathogenesis of MASLD

Increased levels of hepatic cholesterol and triglycerides, elevated serum alanine transaminase and aspartate transaminase, and higher hepatic malondialdehyde levels in rats fed an HFD are indicative of hepatic steatosis and non-alcoholic steatohepatitis (NASH), consistent with our previous findings in this rat model [17]. HFD altered choline metabolism at the transcription and metabolite levels in the pathogenesis of MASLD. In the HFD group, hepatic concentrations of choline metabolites (choline, phosphocholine, and glycerophosphocholine) were higher, while the betaine:choline ratio was lower compared to the ND group. An increasing trend in circulating choline concentrations, along with a decreasing betaine:choline ratio, has been associated with the development and progression of MASLD in epidemiological studies [18,19,20].

HFD altered the hepatic expression of one-carbon metabolism genes in rats, notably downregulating *Mat1a* and *Smpd3*. This downregulation of *Mat1a* correlates with MASLD characteristics in HFD-fed rats, as MAT1A is essential for normal VLDL assembly. In a previous study, *Mat1a*-knockout mice showed decreased VLDL triglyceride secretion and developed MASLD spontaneously [21,22]. Reduced MAT1A and PEMT expression were also observed in patients with NASH and liver fibrosis [23,24]. Wu et al. recently identified SMPD3 methylation as a potential biomarker distinguishing between NASH and mild fibrosis in NAFLD patients, with mild fibrosis associated with hypermethylated SMPD3. The study also found that the *Smpd3* gene in the livers of HFD-induced mice was hypermethylated and suppressed compared to adipose tissue, and a CpG unit within liver *Smpd3* was hypermethylated in mild fibrosis relative to NASH [25]. These findings suggest that the HFD-induced MASLD rats in the current study, whose *Smpd3* was significantly downregulated, were transitioning from NASH to mild fibrosis.

### 3.2. Holy Basil Flower and Fenofibrate Independently Ameliorated Hepatic Insults by Altering Choline Metabolism

Previous studies have shown that *Ocimum sanctum* L. leaf extracts improved serum and hepatic lipid profile, lowered hepatic lipid peroxide content, and increased antioxidant enzyme activity in rats fed a high-cholesterol diet [14]. Our preliminary screening revealed that OSLY may contain flavonoids, terpenoids, alkaloids, fatty acids, fatty amides, benzenes, and organic aromatic compounds (unpublished data) that have been commonly found in *Ocimum* species [12]. In this study, both OSLY and fenofibrate reduced serum triglyceride and LDL cholesterol levels, as well as hepatic cholesterol and triglyceride levels in rats fed HFD. The 1000 mg/kg OSLY, but not fenofibrate, reduced serum cholesterol and increased HDL cholesterol levels compared to the HFD group. OSLY also lowered serum alanine transferase and plasma and hepatic malondialdehyde levels, highlighting its effectiveness in improving serum lipid profile and reducing hepatic injury. No synergistic effects with fenofibrate were found, suggesting that OSLY is better suited as an alternative than an adjunctive therapy.

Fenofibrate independently influenced hepatic choline and one-carbon metabolism, resulting in an increased betaine concentration, a higher betaine:choline ratio, and a reduced GPC concentration compared to the HFD group. This indicates an enhanced supply of methyl groups through choline conversion to betaine in the liver. Fenofibrate also enhanced various parts of the one-carbon metabolism pathway. It may upregulate Mthfd1l to generate formate and increase Pemt expression to utilize methyl groups for PC production. Additionally, fenofibrate may also boost Smpd3 and Chka expression to increase PC production via the CDP-choline pathway. This increased PC availability helps reduce hepatic steatosis in NAFLD rat models by stimulating PPARs and AMP-activated protein kinase [26]. The correlations among genes, metabolites, and metabolic parameters found in this study support the simultaneous activation of the one-carbon pathway to boost PC synthesis and mitigate MASLD/NAFLD pathology. However, the changes in one-carbon enzymes proposed in this study need to be confirmed using appropriate methodologies in future research.

OSLY mildly impacted choline metabolism by reducing hepatic GPC concentrations and increasing *Smpd3* expression. OSLY did not affect other choline metabolites or one-carbon gene transcription.

## 4. Materials and Methods

### 4.1. Holy Basil Flower Extract Preparation

Holy basil flowers were gathered from a community enterprise based in Tambol Mae Sariang, Mae Sariang District, Mae Hong Son, Thailand. A voucher specimen (number 2262) was deposited at the Herbarium Center of Excellence in Agricultural Innovation for Graduate Entrepreneur, Maejo University, Chiang Mai, Thailand. Following collection, the flowers were air-dried, finely ground into a powder, and subjected to extraction using boiling water at a ratio of 100 g per 1 L of distilled water for 1 h. The resulting solution was filtered, and the filtrate was dried using a freeze dryer (Labogene, Lillerød, Denmark). Subsequently, the dried sample underwent further evaporation using a rotary evaporator at 60 °C. The OSLY yield was 16% of the dry flower weight.

### 4.2. Rat Experiments

Adult male Wistar rats (*n* = 48, aged 8 weeks) were obtained from Nomura Siam International Co., Ltd. (Bangkok, Thailand). The animals were kept in a controlled environment (25 ± 1 °C and 70% humidity) with a 12 h dark/light cycle. At the beginning of the experiment (week 0), rats were randomly divided into eight experimental groups (6 rats per group) and fed either a normal diet (ND; 2 groups) or a high-fat diet (HFD; 6 groups) ad libitum for 12 weeks (Figure 7). We previously showed that this duration is sufficient for obesity, insulin resistance, and steatosis to develop [17]. The ND was a commercial chow diet (C.P. Mice Feed Food No. 082, Bangkok, Thailand); 20% of the diet’s calories were fat, 28% were protein, and 52% were carbohydrates. The HFD was prepared in-house such that 60% of its calories were fat, 26% were protein, and 14% were carbohydrates, as previously described in Kaewmalee et al. (2021) [27]. At the end of the 12 weeks, increased body weight, total cholesterol, and the homeostasis model of insulin resistance index were used to confirm insulin resistance in HFD-fed rats compared to ND-fed rats.

From week 13 to week 24, ND-fed rats were randomly assigned to receive either 1000 mg/kg body weight (bw) OSLY (*n* = 6) or a vehicle (distilled water; *n* = 6) by daily oral gavage. HFD-fed rats were randomly divided into six groups (*n* = 6 each) according to treatment: 250, 500, or 1000 mg/kg bw OSLY, 50 mg/kg bw fenofibrate (an approved anti-dyslipidemic drug), fenofibrate + 1000 mg/kg bw OSLY, or a vehicle (distilled water, negative control). The dosages of OSLY used in the current study were comparable to the holy basil leaf extract dosages (590 mg/kg bw) that were shown to decrease hepatic lipid content in a previous study [14]. Therefore, the 250, 500, and 1000 mg/kg bw doses were chosen to investigate whether lower, similar, or higher doses of OSLY would yield different results compared to the holy basil leaves. The human equivalent dose for 1000 mg/kg bw was 1.5 g for a 70 kg person [28]. Additionally, the OSLY doses contained total polyphenol amounts in a similar range as the amounts that prevented steatohepatitis in this rat model in our previous study [17].

Amounts of food and water intake were recorded weekly throughout the experimental duration. At the end of the experiment, fasted rats were sacrificed to harvest blood and liver samples. The study protocol received approval from the Laboratory Animal Care and Use Committee at the Faculty of Medicine, Chiang Mai University, Chiang Mai, Thailand (protocol numbers: 05/2565 and 40/2566).

### 4.3. Sample Collection and Processing

Rats were fasted for 6–8 h before being intraperitoneally injected with 40–50 mg/kg bw thiopental. The blood samples were collected directly by cardiac puncture and centrifuged at 12,000× *g* for 1 min to separate plasma and blood cells. Plasma samples were stored at −20 °C. The liver was collected, snap frozen in liquid nitrogen, and stored at −80 °C. Small amounts of liver samples were cryo-homogenized using liquid nitrogen for analysis of choline metabolites.

### 4.4. Analytic Measurements

#### 4.4.1. Blood and Liver Biochemical Assays

Plasma biochemical parameters, including glucose, aspartate aminotransferase, alanine aminotransferase, alkaline phosphatase, blood urea nitrogen, and creatinine, were measured using commercial assay kits (Merck). The estimated glomerular filtration rate (eGFR) was calculated using the following formula: eGFR (μL/min) = 5862 × body weight^0.695^ × creatinine(μM)^−1.15^ × urea(mM)^−0.391^ [29]. Plasma cholesterol and triglycerides were determined using a commercial assay kit (Biotechnical Co., Ltd., Bangkok, Thailand), with absorbance measured at 500 nm. HDL levels were measured by using a commercial assay kit (HUMAN GmbH, Wiesbaden, Germany). LDL levels were calculated from the following formula: LDL (mg/dL) = total cholesterol − HDL − (triglycerides/5). Lipid content was extracted from liver samples using 2:1 (*v*/*v*) chloroform/methanol followed by centrifugation. Cholesterol and triglycerides in the supernatant were measured using a commercial assay kit (Biotechnical Co., Ltd., Bangkok, Thailand). Malondialdehyde levels were measured using a commercial assay kit (Cayman Chemical, Ann Arbor, MI, USA).

#### 4.4.2. Measurements of Hepatic Choline Metabolites

Aqueous (choline, phosphocholine, glycerophosphocholine (GPC), and betaine) and organic choline metabolites (PC, lysoPC, and sphingomyelin) were measured in liver samples using stable isotope dilution LC-MS/MS according to the methods used by Koc et al. [30] with modification based on our instrument. Specifically, 0.02 g of sample was mixed with 400 µL methanol/chloroform 2:1 *v*/*v* and kept at −20 °C overnight (15 h). The next day, the internal standard mix was added, and the tubes were vortexed and centrifuged at 13,000 rpm 4 °C for 5 min. The supernatant was poured out into a new tube. Then, 250 µL of 2:1:0.8 (*v*/*v*/*v*) methanol/chloroform/water was added to the residues. The tubes were vortexed and centrifuged at 13,000 rpm 4 °C for 5 min. The supernatant was combined with the first extraction. Next, 100 µL of chloroform and 120 µL of water were added to the tube of supernatant. The tubes were vortexed and centrifuged at 13,000 rpm 4 °C for 5 min. At this point, the solution formed an aqueous phase (top layer) and an organic phase (bottom layer) with a white layer of protein in the middle. The bottom layer was drawn out into a new tube, while the protein layer was discarded. Subsequently, the two phases were dried using a vacuum concentrator (Labconco Corporation, Kansas City, MO, USA).

After drying, the organic phase was redissolved in 500 µL hexane:chloroform: methanol (95:3:2). Bond Elute Aminopropyl columns (Agilent Technologies, Santa Clara, CA, USA) were conditioned with 5 mL hexane and loaded with the redissolved samples. Then, the columns were washed with 4 mL chloroform and 4 mL 2% acetic acid in ethyl-ether, respectively. The compounds were eluted into 2 mL cryogenic tubes by loading 1 mL of methanol twice. A measure of 500 µL of the eluent was dried in the vacuum concentrator to remove residual chloroform and resuspended in 500 µL methanol. A further 10 µL of the eluent was injected into the LC-MS/MS. The dried aqueous phase was resuspended for 15 min in 20 µL of water. Next, 100 µL of methanol was added, and the solution was sonicated for 5 min, followed by centrifugation at 13,000 rpm 4 °C for 5 min. A measure of 10 µL of the supernatant was injected into the LC-MS/MS.

Choline metabolites were quantified using TSQ Quantis Ion Triple Quadrupole MS/MS with Ultimate 3000 UHPLC system (Thermo Fisher Scientific, Waltham, MA, USA). Compounds were separated onto a Syncronis Silica column (150 mm × 2.1 mm, 5 µ; Thermo Fisher Scientific, USA). Mobile phases were 3 mM ammonium acetate in acetonitrile (buffer A: 800 mL acetonitrile, 127 mL water, 68 mL methanol, 3 mL 1M ammonium acetate, and 2 mL acetic acid) and 2.4 mM ammonium acetate in acetonitrile (buffer B: 500 mL acetonitrile, 500 mL water, 85 mL methanol, 27 mL 1M ammonium acetate, 18 mL acetic acid). Aqueous choline metabolites were separated using a 30 min HPLC gradient: 0 min, 0% B at 0.4 mL/min; 2 min, 0% B at 0.4 mL/min; 11 min, 45% B at 0.4 mL/min; 14 min, 45% B at 0.4 mL/min; 18 min, 100% B at 0.3 mL/min; 21.5 min, 100% B at 0.5 mL/min; 25.5 min, 0% B at 0.4 mL/min; 30 min, 0% B at 0.4 mL/min. Organic choline metabolites were separated using a 15 min HPLC gradient using a constant flow rate of 0.4 mL/min: 0 min, 5% B; 3 min, 5% B; 4.5 min, 37% B; 5.5 min, 53% B; 6.5 min, 100% B; 9.5 min, 100% B; 9.6 min, 5% B; 15 min, 5% B. Ions were generated in the MS/MS using heated electrospray ionization and detected using selected reaction monitoring in positive mode with the following m/z ratios: choline, 104/60; d13-choline, 117/69; betaine, 118/58; d9-betaine, 127/68; GPC 258/104; d9-GPC, 267/113; phosphocholine, 184/124; d9-phosphocholine 193/125. The lipid-soluble choline metabolites underwent in-source fragmentation to generate phosphocholine (m/z 184) and d9-phosphocholine (m/z 193), which were detected using selected ion monitoring.

All data were processed using Chromeleon Software 7.3.2 (Thermo Fisher Scientific, USA). Total concentrations were calculated using the peak area of analyte/peak area of internal standard, plotted against standard curves (R^2^ > 0.99). The coefficient of variation of the controls was 1.5–19% for aqueous metabolites and 2–17% for organic metabolites.

#### 4.4.3. Hepatic One-Carbon Metabolism Gene Expression

Total RNA was extracted from liver tissues using TRIzol™ Reagent (Thermo Fisher Scientific, Waltham, MA, USA) and reverse transcribed into first-strand cDNA using a commercial first-strand cDNA kit (Bio-Rad Laboratories, Hercules, CA, USA). Quantitative real-time PCR was performed using SYBR green real-time PCR master mix (Bioline, London, UK) in an ABI 7500 instrument (Life Technologies, Grand Island, NY, USA). The primer sets are shown in Table 1 (Macrogen, Seoul, Republic of Korea). mRNA levels were calculated by the ΔCt method, normalized to the expression of reference gene β-actin.

### 4.5. Statistical Analysis

The effects of treatments on hepatic choline metabolite concentrations and one-carbon gene expression were analyzed using one-way ANOVA with the Dunnett post hoc test. Associations between choline metabolites, metabolic parameters, and gene expression were investigated using correlation analysis with false discovery rate adjustment. Principal component analysis followed by multiple linear regression models was used to explore the influence of choline metabolites on metabolic parameters. Data were expressed as medians and interquartile ranges. All analysis was conducted using R (ver 4.4.0) at a significance of *p* < 0.05.

## 5. Conclusions

Rats with high-fat-diet-induced obesity and signs of MASLD show alteration in hepatic choline metabolites and one-carbon gene transcription, implicating choline/one-carbon metabolism in the pathogenesis of MASLD. A comparison of holy basil flower (*Ocimum sanctum* L.) extracts with fenofibrate in these rats shows that the holy basil flower extracts improve serum lipid profile, reduce hepatic lipid accumulation, and alleviate hepatic injury and oxidation. These effects of the holy basil flower extracts are as potent as or more effective than fenofibrate. However, combining both treatments does not yield any synergistic effect, indicating that the holy basil flower extracts merit further exploration as an alternative medicine. Additionally, the lipid-lowering action of fenofibrate is partially influenced by choline metabolism, although this pathway is only slightly affected by the holy basil flower extracts.

## Figures and Tables

**Figure 1 plants-14-00013-f001:**
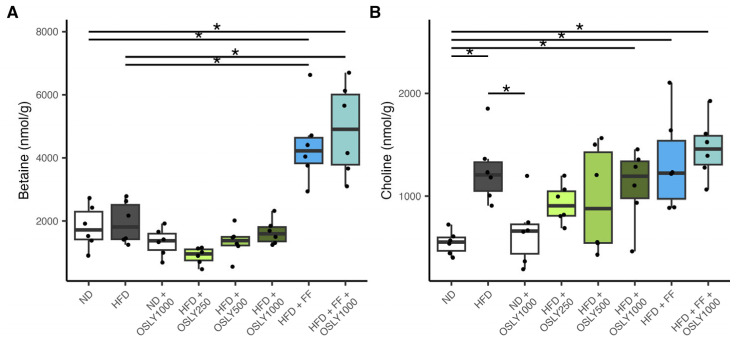
Hepatic content of aqueous choline metabolites (betaine (**A**), choline (**B**), betaine:choline ratio (**C**), phosphocholine (**D**), and glycerophosphocholine (**E**)) in rats fed either a normal diet (ND) or a high-fat diet (HFD). The rats were administered a vehicle (distilled water), *Ocimum sanctum* L. flower extracts (OSLY) at 250, 500, or 1000 mg/kg body weight, and/or fenofibrate (FF) daily for 12 weeks. * Indicates significant difference (*p* < 0.05) based on ANOVA with Dunnett post-hoc test.

**Figure 2 plants-14-00013-f002:**
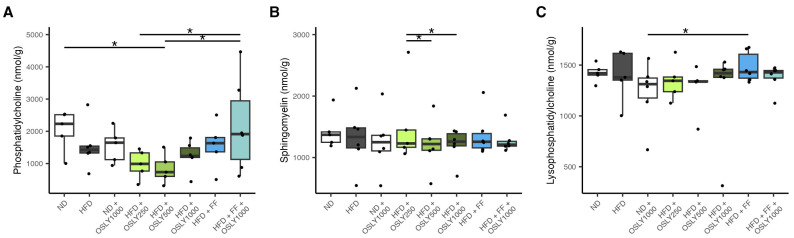
Hepatic content of lipid-soluble choline metabolites (phosphatidylcholine (**A**), sphingomyelin (**B**), and lysophosphatidylcholine (**C**)) in rats fed either a normal diet (ND) or a high-fat diet (HFD). The rats were administered a vehicle (distilled water), *Ocimum sanctum* L. flower extracts (OSLY) at 250, 500, or 1000 mg/kg body weight, and/or fenofibrate (FF) daily for 12 weeks. * Indicates significant difference (*p* < 0.05) based on ANOVA with Dunnett post-hoc test.

**Figure 3 plants-14-00013-f003:**
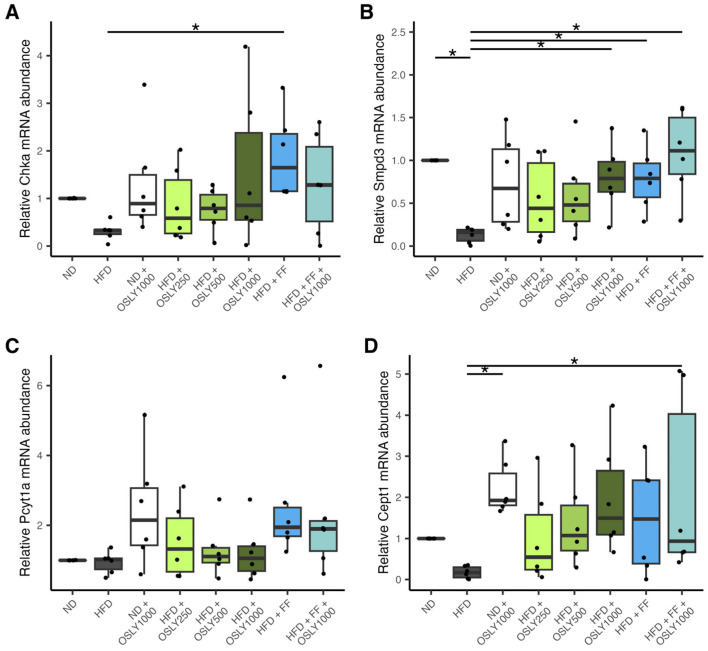
Relative mRNA abundance of CDP-choline pathway genes in rat liver—(**A**) choline kinase alpha (*Chka*); (**B**) sphingomyelin phosphodiesterase 3 (*Smpd3*); (**C**) phosphate cytidylyltransferase 1α isoform (*Pcyt1a*); (**D**) choline-ethanolamine phosphotransferase-1 (*Cept1*) in rats fed either a normal diet (ND) or a high-fat diet (HFD). The rats were administered a vehicle (distilled water), *Ocimum sanctum* L. flower extracts (OSLY) at 250, 500, or 1000 mg/kg body weight, and/or fenofibrate (FF) daily for 12 weeks. * Indicates significant difference (*p* < 0.05) based on ANOVA with Dunnett post-hoc test.

**Figure 4 plants-14-00013-f004:**
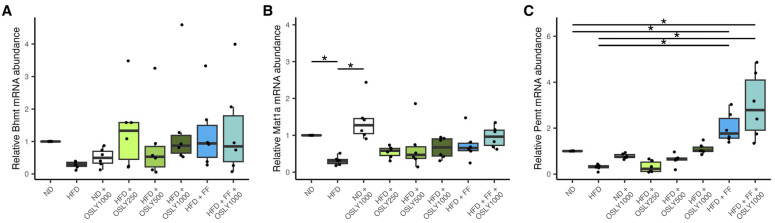
Relative mRNA abundance of methionine cycle genes in rat liver—(**A**) betaine-homocysteine methyltransferase (*Bhmt*); (**B**) methionine adenosyltransferase 1A (*Mat1a*); (**C**) phosphatidylethanolamine N-methyltransferase (*Pemt*) in rats fed either a normal diet (ND) or a high-fat diet (HFD). The rats were administered a vehicle (distilled water), *Ocimum sanctum* L. flower extracts (OSLY) at 250, 500, or 1000 mg/kg body weight, and/or fenofibrate (FF) daily for 12 weeks. * Indicates a significant difference (*p* < 0.05) based on ANOVA with Dunnett post-hoc test.

**Figure 5 plants-14-00013-f005:**
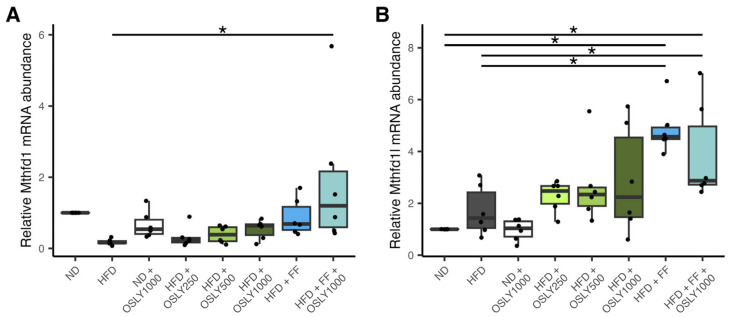
Relative mRNA abundance of folate cycle genes in rat liver—(**A**) methylenetetrahydrofolate dehydrogenase 1 (*Mthfd1*, cytoplasmic form); (**B**) Mthfd1−like gene (*Mthfd1l*, mitochondrial form) in rats fed either a normal diet (ND) or a high-fat diet (HFD). The rats were administered a vehicle (distilled water), *Ocimum sanctum* L. flower extracts (OSLY) at 250, 500, or 1000 mg/kg body weight, and/or fenofibrate (FF) daily for 12 weeks. * Indicates significant difference (*p* < 0.05) based on ANOVA with Dunnett post-hoc test.

**Figure 6 plants-14-00013-f006:**
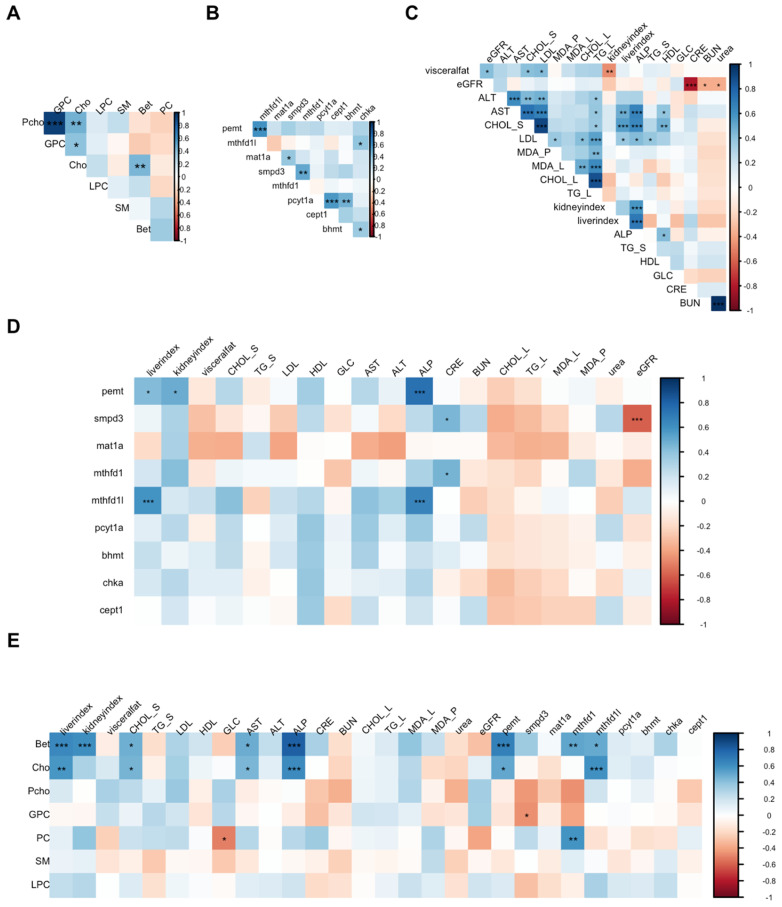
Correlation within choline metabolites (**A**), one-carbon metabolism gene expression (**B**), biochemical parameters (**C**), and between gene expression, metabolites, and biochemical parameters (**D**,**E**). Blue and red colors indicate positive and negative correlations, respectively. Darker colors indicate stronger correlations in either direction. * Indicates significant correlation at *p* < 0.05, ** at *p* < 0.01, and *** at *p* < 0.001 after adjusting for false discovery rate. Choline metabolites—Bet, betaine; Cho, choline; GPC, glycerophosphocholine; LPC, lysophosphatidylcholine; PC, phosphatidylcholine; Pcho, phosphocholine; SM, sphingomyelin. One-carbon metabolism genes—*Bhmt*, betaine-homocysteine methyltransferase; *Cept1*, choline-ethanolamine phosphotransferase−1; *Chka*, choline kinase alpha; *Mat1a*, methionine adenosyltransferase 1A; *Mthfd1*, methylenetetrahydrofolate dehydrogenase 1; *Mthfd1l*, Mtfhd1−like gene; *Pcyt1a*, phosphate cytidylyltransferase 1 choline α isoform; *Pemt*, phosphatidylethanolamine N-methyltransferase; *Smpd3*, sphingomyelin phosphodiesterase 3. Biochemical markers—ALT, alanine transaminase; ALP, alkaline phosphatase; AST, aspartate transaminase; BUN, blood urea nitrogen; CHOL_L, liver cholesterol; CHOL_S, serum cholesterol; CRE, creatinine; eGFR, estimated glomerular filtration rate; GLC, blood glucose; HDL, high-density lipoprotein; LDL, low-density lipoprotein; MDA_L, liver malondialdehyde; MDA_P, plasma malondialdehyde; TG_L, liver triglyceride; TG_S, serum triglyceride.

**Figure 7 plants-14-00013-f007:**
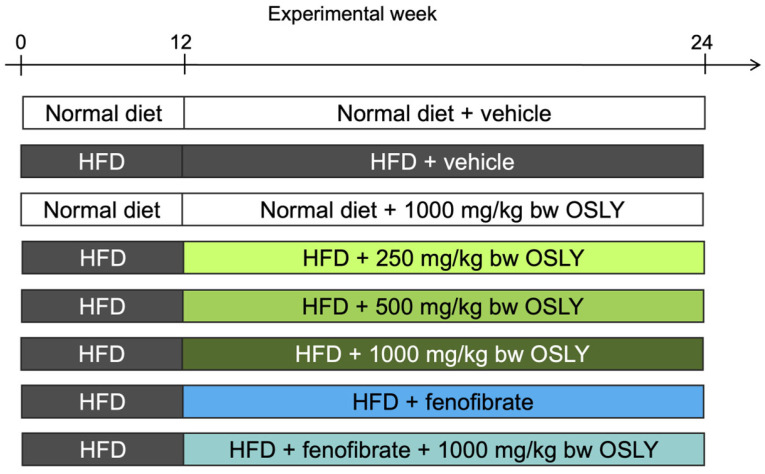
Experimental design. Adult male Wistar rats were fed either a normal or high-fat diet (HFD) for 12 weeks. Then, rats were administered either a vehicle (distilled water), *Ocimum sanctum* L. flower extract (OSLY) at 250, 500 or 1000 mg/kg body weight (bw), fenofibrate, or fenofibrate + 1000 mg/kg bw OSLY daily for 12 weeks. Blood and liver samples were collected at the end for analysis of choline metabolites and metabolic parameters.

**Table 1 plants-14-00013-t001:** Rat primer sequences and amplicon sizes for one carbon metabolism gene amplification.

Gene Symbols	Gene Name	Forward Primers 5′ to 3′	Reverse Primers 3′ to 5′	RT-PCR Product Size (bp)
*Pemt*	Phosphatidylethanolamine N-methyltransferase	CCAGGTTGCACAAAAGGAGC	GTCAGGGATTGGTGGGGATG	198
*Mat1a*	Methionine adenosyltransferase 1A	AGCCTGGGTGTGTCTGTCTA	TCGATATGAGCCAGGTCCGT	165
*Mthfd1*	Methylenetetrahydrofolate dehydrogenase 1	GCGGCTATTCCCAGGTCATT	ATAGCAGCAGCCACAAGGTT	100
*Mtfhfd1l*	Mitochondrial monofunctional 10-formyl-tetrahydrofolate synthetase	TGCCGAGGGACTTCATTCTG	ACCTGGCATTGTGCTCATCA	90
*Cept1*	Choline-ethanolamine phosphotransferase-1	TGGGCTGGACATAACTGGGTA	GGCTATTTACCACGCAGGC	170
*Pcyt1a*	Phosphate cytidylyltransferase 1, choline, α isoform	CGTCTCCCCGCAACCTATTT	TGTTGCTCCATTAGGGCCAG	172
*Bhmt*	Betaine-homocysteine methyltransferase	AAGCCTTTGCTGGAGACCG	ATCTCCGATCACGACTTCGC	147
*Chka*	Choline kinase alpha	GTCTCTCGTCACTGCTGCTC	GAATGGCTCACCGGCTTCA	118
*Smpd3*	Sphingomyelin phosphodiesterase 3	TATGGCAGCTTGGCACTAGG	AGATTCCTGGGAGGTCAGGC	161
*β-Actin*	β-Actin	CCTAAGGCCAACCGTGAAAA	GGAGCGCGTAACCCTCAATAC	189

## Data Availability

The original contributions presented in the study are included in the article.

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
