# Peer review of "Holy Basil (Ocimum sanctum L.) Flower and Fenofibrate Improve Lipid Profiles in Rats with Metabolic Dysfunction Associated Steatotic Liver Disease (MASLD): The Role of Choline Metabolism"

_plants, 2024, doi:10.3390/plants14010013_

Round 1

Reviewer 1 Report

Comments and Suggestions for Authors

Dear Authors,

I am delighted very much to review an interesting article. The authors demonstrated that fenofibrate and OSLY independently improve lipid profiles in MASLD rats. The benefits of fenofibrate are partially mediated by choline/one-carbon metabolism, while those of OSLY are not mediated by this pathway. However, there are some comments and suggestions that the authors may want to consider.

1.       The compositional differences between Ocimum sanctum L. flower extract (OSLY) and leaf extract should be elaborated in the discussion, highlighting the variations in their components and respective effects.

2.     The rationale for choosing fenofibrate as a comparator with OSLY should be clearly explained in either the introduction or the discussion.

3.     There seems to be a potential discrepancy between the title and the conclusions. While the title suggests that both treatments improve lipid profiles by altering choline metabolism, the conclusion states that OSLY's effects are not mediated by this pathway. This inconsistency should be addressed to ensure alignment.

4.     The study should clarify whether the observed gene expression changes following OSLY treatment follow a dose-dependent relationship.

5.     The detection methods used in the study are somewhat limited; for example, Smpd3 was only analyzed using PCR. Including additional validation methods, such as Western blotting or immunohistochemistry, would significantly enhance the credibility and robustness of the findings.

Comments on the Quality of English Language

The manuscript requires language refinement to improve readability and ensure clarity for a wider audience.

Reviewer 2 Report

Comments and Suggestions for Authors

Dear Authors,

Please the rationale behind the work must be clear, why do you hypothesize that holy basil flowers could alter choline metabolites and one-carbon metabolism gene expression? Please support your arguments in the literature.

 Who identifies the plant? Voucher number?

 Please fully describe the HFD preparation.

 Please improve the quality of figure 7.

 Please explain how you select the administered doses.

 Figures 1 to 6 are of bad quality and cannot be interpreted.

 Exactly where is the contribution to plant science?

Unfortunately, some results are illegible (figures) so in the new version I need to review the results.

Reviewer 3 Report

Comments and Suggestions for Authors

The manuscript bellow is suitable for the publication in the journal Plants, after minor changes will be made.  The research carried out is well structured and described with many figures and tables, the methods are complete described and the conclusions support the studies.

This study is based on the phytochemical and pharmacological characterization of the extract from the flowers of Ocimum sanctum L., but it is not specified if new compounds were identified after the phytochemical analysis.

Specific comments:

-        Line 66 – the family of the species must be included

-        In chapter 3.1 please specify if some compounds found in this water extract of the flowers have been identified in this species (but in other parts of the plant) or in other species of the genus  Ocimum.   

Round 2

Reviewer 2 Report

Comments and Suggestions for Authors

Dear Authors, many thanks for you answers, please the contribution to plant science must be reflected in the manuscript.
The used doses is a major concern, 1000 mg/Kg is equivalent to give 70 g of the extract to a 70 K person!
Please add the extraction yield in the results.

Round 3

Reviewer 2 Report

Comments and Suggestions for Authors

Dear authors, thanks for your answers, yes I am aware how the rat/human doses is calculated, my concern is because in the ms is not clear how do you calculate the doses, please add the reference in section 4.2

Author Response

Comment 1: Dear authors, thanks for your answers, yes I am aware how the rat/human doses is calculated, my concern is because in the ms is not clear how do you calculate the doses, please add the reference in section 4.2

Response: Thank you again for providing inputs on our manuscript. Per you suggestion, we added the reference and the following sentence in Lines 382-383: "The human equivalent dose for 1,000 mg/kg bw was 1.5 g for a 70 kg person [39]."